# Genetic Differentiation of the Bloodsucking Midge *Forcipomyia taiwana* (Diptera: Ceratopogonidae): Implication of the Geographic Isolation by the Central Mountain Ranges in Taiwan

**DOI:** 10.3390/insects15010023

**Published:** 2024-01-01

**Authors:** Yung-Hao Ching, Yuan-Chen Kuo, Ming-Ching Su, Szu-Chieh Wang, Chuen-Fu Lin, Wu-Chun Tu, Ming-Der Lin

**Affiliations:** 1Department of Molecular Biology and Human Genetics, Tzu Chi University, Hualien 97004, Taiwan; yching@gms.tcu.edu.tw (Y.-H.C.); vincent890203@gmail.com (Y.-C.K.); jingg888@gmail.com (M.-C.S.); scwang2023@gmail.com (S.-C.W.); 2Department of Veterinary Medicine, College of Veterinary Medicine, National Pingtung University of Science and Technology, Pingtung 91201, Taiwan; cflin2283@mail.npust.edu.tw; 3Department of Entomology, National Chung Hsing University, Taichung 40227, Taiwan; wctu@dragon.nchu.edu.tw; 4Institute of Medical Science, Tzu Chi University, Hualien 97004, Taiwan

**Keywords:** biting midge, *Forcipomyia*, microsatellite marker, gene flow, geographical isolation

## Abstract

**Simple Summary:**

Our study elucidates a genetic differentiation of *F. taiwana* populations across the central mountain ranges of Taiwan, which stands out as a significant geographical feature shaping these populations’ genetic landscape. The mountain range acts as a significant topographic barrier, influencing gene flow and contributing to distinct genetic clusters in eastern and western regions. Despite potential mechanisms for passive dispersal, such as human-mediated transportation and monsoon winds, the genetic divergence remains pronounced. These findings provide crucial insights for understanding the species’ molecular ecology and can inform targeted pest management strategies. Future research, involving more detailed genomic analyses, will be pivotal in exploring specific genetic adaptations and further elucidating gene flow across these geographical divisions.

**Abstract:**

*Forcipomyia* (*Lasiohelea*) *taiwana*, a small bloodsucking midge, thrives in moderately moist habitats and is commonly found in grassy and bushy areas at an elevation below 250 m. This species exhibits a diurnal biting pattern and shows a marked preference for human blood. Although not known to transmit arthropod-borne diseases, the bites of *F. taiwana* can induce severe allergic reactions in some individuals. As a significant nuisance in Taiwan, affecting both daily life and the tourism industry, comprehensive studies on its population genetics across different geographical regions remain scarce. The central mountain ranges in Taiwan, comprising more than two hundred peaks above 3000 m in elevation, extend from the north to the south of the island, creating distinct eastern and western geographical divisions. This study utilizes microsatellite markers to explore the genetic differentiation of *F. taiwana* populations located in the eastern and western regions of the mountain ranges. Our findings reveal substantial genetic differentiation among populations inhabiting Taiwan’s western region compared to those in the eastern region. This indicates that the topographical barriers presented by the mountain ranges significantly restrict gene flow, particularly given the species’ limited active flight ability and habitat preferences. Although passive dispersal mechanisms, like wind or human activity, could contribute, this study concludes that the gene flow of *F. taiwana* between the western and eastern regions is primarily influenced by topographical constraints.

## 1. Introduction

*Forcipomyia* (*Lasiohelea*) *taiwana* (Shiraki) (Diptera: Ceratopogonidae), a minuscule bloodsucking midge of about 1.4 mm in length, is commonly found in the urban and suburban locales of Taiwan, especially during the spring and summer months [1]. Known for its marked predilection for human blood [2], the species exhibits diurnal biting behavior, typically starting around two hours after sunrise [3]. For females of this species, obtaining a blood meal is crucial for ovarian development and successful reproduction [4]. Although *F. taiwana* does not transmit arthropod-borne viruses, its bites may induce severe allergic reactions in certain individuals [5]. Additionally, large swarms of these midges can considerably disrupt outdoor activities, thereby adversely affecting Taiwan’s leisure sector [1].

Initially discovered in 1913 by Tokuichi Shiraki in Taichung County, Taiwan [6], *F. taiwana* was later identified across 11 Taiwanese counties, except Miaoli, Kaohsiung, Pingtung, and Taitung, between 1990 and 1991 [1]. Subsequent surveys from 2009 to 2012 revealed its presence throughout the island [3,7]. Thriving in moderately moist environments, *F. taiwana* is typically found in grassy, bush-rich areas, such as parks, school grounds, and villages near foothills and recreational sites [8]. Its altitudinal distribution mainly falls below 250 m, with occurrences up to 500 m [9]. The prevalence of this species is closely linked to human activity, particularly due to the female’s strong preference for human blood. The females exhibit limited flight capabilities, averaging around 2 m in height and rarely exceeding 4 m [8,10]. Their restricted straight flight ability suggests that walking at speeds over 1 m per second can effectively prevent bites, indicating that *F. taiwana* might not engage in active host-seeking behavior and that their egg-laying sites may coincide with adult habitats [8]. Breeding year-round, female *F. taiwana* typically deposit eggs in sunlit, humid soil, where blue-green or green algae, essential for larval nutrition, are abundant [8,11,12]. Following a blood meal, a gravid female lays approximately 40 eggs over two to three days [4,11,13]. The life cycle, spanning approximately 16 days in conditions of 26 °C and 70% relative humidity, includes embryogenesis (2–3 days), four larval stages (7–10 days), and pupal development (3 days) [13]. Hence, in environments with abundant human blood sources, *F. taiwana* populations can swiftly multiply.

While *F. taiwana*’s widespread presence is evident, the population genetics and genetic structure underlying its distribution remain poorly understood. To unravel the genetic underpinnings of *F. taiwana*’s distribution pattern, we employed microsatellite markers. Microsatellites, also known as simple sequence repeats or short tandem repeats, consist of 1–6 bp nucleotide repeats and can be found throughout eukaryotic nuclear genomes. They exhibit co-dominance, high polymorphism, follow Mendelian inheritance patterns, and are often selectively neutral in non-coding regions. In this study, we adopted PacBio long-read sequencing technology to develop microsatellite markers. This approach streamlined the process and significantly enhanced the efficiency of microsatellite marker development. We developed five sets of microsatellite markers and sampled 651 female *F. taiwana* specimens from eleven different geographic regions, situated on both sides of Taiwan’s central mountain ranges. Our results revealed considerable genetic differentiation between the populations on the western side compared to those on the eastern side of the range. This suggests that the topographical barriers posed by the mountain ranges might significantly impede gene flow within these populations. Such findings highlight that the dispersal of *F. taiwana* is intricately linked to the landscape, and that factors like wind or human-assisted spread, as potential passive dispersal mechanisms, warrant further investigation for a comprehensive understanding of *F. taiwana*’s dispersion mechanism.

## 2. Materials and Methods

### 2.1. Sample Collection 

In this study, 651 female *F. taiwana* specimens were collected from eleven different geographic locations in the eastern and western regions of the central mountain ranges (Figure 1). The identification of these specimens was conducted following the guidelines outlined in the ‘Catalogue and Keys of Chinese Ceratopogonidae (Insecta, Diptera)’ [14]. For the purpose of our study, a group of *F. taiwana* specimens collected from a specific geographic location was defined as a “population” (refer to Table 1 for detailed locations and sample sizes). To ensure the preservation of the genetic material, all collected specimens were immediately stored in 95% ethanol. Subsequent storage was performed at 4 °C to maintain the integrity of the samples until further analysis could be conducted. This approach was designed to optimize the quality of the genetic material for the subsequent development of microsatellite markers and detailed genetic analysis.

### 2.2. Isolation of Genomic DNA

For the extraction of polymerase chain reaction (PCR)-ready genomic DNA from *F. taiwana* specimens, we employed a modified version of the HotSHOT method [15]. The process began with the homogenization of each individual midge in 550 μL of 50 mM NaOH. This mixture was then heated to 95 °C for 20 min. Following this, the samples were rapidly cooled on ice for 5 min. After the initial lysis and cooling, the DNA solution was neutralized by adding 50 μL of 1M Tris-HCl (pH 8.0). The sample was then centrifuged at 16,000× g for 10 min at room temperature. The clear supernatant, containing the genomic DNA, can be kept at 4 °C for several weeks without significant degradation. For long-term preservation, the supernatant can be stored at −20 °C.

### 2.3. Amplification of Microsatellite Markers and Fragment Analysis

For the development of microsatellite markers in *F. taiwana*, we utilized Circular Consensus Sequencing reads (GenBank accession No. MW921469 to MW921473) generated from the PacBio Sequel System (Pacific Biosciences, Menlo Park, CA, USA). The primer design for PCR was facilitated using Primer 3 software using default settings [16], as detailed in Table 2. Following the method developed by Oetting et al. [17], with some modifications, each forward primer includes a 19 bp *M13* sequence (CACGACGTTGTAAAACGAC) at its 5′ end. We employed fluorescent-labeled *M13* primers (Life Technologies, Carlsbad, CA, USA) in the PCR reaction, enabling the generation of fluorescent-labeled fragments suitable for fragment analysis. The specific fluorescent labels used for different microsatellite markers were PET for *P1*, VIC for *P2* and *P4*, and 6-FAM for *P3* and *P5*. The PCR reactions were conducted in a 10 μL mixture, consisting of 5 μL of *Taq* master mix (amaR OnePCR from GeneDireX, Taoyuan, Taiwan), 1 μL of PCR-ready genomic DNA, 30 nM of the fluorescent-labeled *M13* primer, and varying amounts of the specific forward and reverse primers (refer to Table 2 for details). The PCR conditions included an initial denaturation at 95 °C for 30 s, followed by 5 cycles of 95 °C for 30 s, 60 °C for 30 s, and 72 °C for 30 s. This was followed by 30 cycles at 95 °C for 30 s, 50 °C for 30 s, and 72 °C for 30 s, concluding with a final extension at 72 °C for 40 min. The PCR products were processed on the LabCycler 96 (SensoQuest, Göttingen, Germany). For fragment analysis, the fluorescent-labeled PCR products of the five microsatellite markers were pooled in equal volumes and analyzed using an ABI 3730XL DNA Analyzer (Applied Biosystems, Bedford, MA, USA). GeneScan 500 LIZ was employed as a size standard in this analysis.

### 2.4. Genotype Scoring and Data Analysis

For the analysis of microsatellite data, we used Geneious software (version R11) (Biomatters Ltd., Auckland, New Zealand), equipped with the Microsatellite Plugin (version 1.4.7), to conduct size analysis and peak calling. The data obtained from fragment analysis were subjected to population genetic analysis using GenAlEx (ver. 6.51b2) [18,19]. We performed frequency-based analyses, including the calculation of the number of effective alleles (*Ne*), allele frequencies, observed heterozygosities (*Ho*), expected heterozygosities (*He*), and the inbreeding coefficient/fixation index (*F*). Analysis of Molecular Variance (AMOVA) was used to assess pairwise genetic differentiation (*F_ST_*) with 999 permutations. For visualizing the hierarchical clustered heatmap based on pairwise *F_ST_* values derived from AMOVA, we employed the pheatmap package (version 1.0.12) in R. Both AMOVA and Principal Coordinate Analysis (PCoA) were performed using the GenAlEx software. To identify the genetic structure of the sampled populations, we utilized the Bayesian clustering algorithm implemented in the software STRUCTURE (ver. 2.3.4) [20]. We ran the analysis under an admixture model with correlated allele frequencies. For each potential number of clusters (K), ranging from one to five, we conducted 50 independent runs to ensure consistency across results. Each run consisted of a burn-in period of 100,000 iterations followed by 100,000 Markov Chain Monte Carlo (MCMC) replicates. The optimal number of clusters was determined using the ΔK method [21], as implemented in Structure Harvester [22]. This approach assesses the rate of change in the log probability of data between successive K values to identify the most likely number of genetic clusters. Following the determination of the optimal K, we utilized Clumpp (ver. 1.1.2) software [23] to align multiple runs and average the membership coefficients for each individual across all runs for the best K. This step was crucial to account for label switching and multimodality issues inherent in STRUCTURE outputs. We selected the Greedy algorithm with 1000 random input orders in Clumpp to achieve the optimal alignment of individual assignments across different runs.

## 3. Results

### 3.1. Genetic Structures of F. taiwana Populations Revealed by Microsatellite Analysis

Taiwan is geographically divided into distinct eastern and western regions by a series of mountain ranges, which run through the heart of the island from the north to the south (Figure 1). These are referred to as the central mountain ranges in this manuscript. The western populations of *F. taiwana* are separated from their eastern counterparts by the mountain ranges, which serve as a significant geographical barrier with an average elevation of 2000 m. To determine the genetic structure of *F. taiwana* populations on either side of the mountain ranges, we sampled female *F. taiwana* specimens from eleven geographical locations (Figure 1). Specifically, in the eastern region, 333 specimens were collected from seven locations, including Hualien County (HL) and Yilan County (YL). In the western region, 189 specimens were gathered from Taichung City (TC) and Changhua County (CH), and 129 specimens from Pingtung County (PT) and Kaohsiung City (KH) in the southwestern region (Table 1).

We designed five microsatellite markers for *F. taiwana*, utilizing circular consensus sequencing reads from PacBio long-read sequencing. A total of 651 samples from eleven locations, referred to as populations in this study, were genotyped using these markers (Table 2). All markers were polymorphic, with fragment sizes ranging from 138 to 465 bp, and the number of alleles per locus varied from 5 to 8. Missing data were minimal, observed as only 0.15% in *P1* and *P2*. Analysis via Micro-Checker software [24] indicated no evidence of null alleles across all examined populations.

The heterozygosity analysis (*Ho*, *He*, and *uHe*) of the eleven *F. taiwana* populations, detailed in Table 3, reveals diverse genetic profiles. Some populations displayed high levels of both observed (*Ho*) and expected (*He*) heterozygosity, indicative of significant genetic diversity and effective random mating. In contrast, discrepancies between *Ho* and *He* in other populations suggest potential inbreeding or population substructuring. For example, in the TC_DKMZ population, the alignment of *Ho* (0.589), *He* (0.566), and *uHe* (0.569) for locus *P1* demonstrated a good alignment of observed and expected heterozygosity, indicative of a rich genetic variation and random mating. However, the HL_RRRS population showed a lower *Ho* (0.106) compared to *He* (0.173) for locus *P3*, hinting at genetic uniformity and possible inbreeding, yet other loci in the same population (*P1*, *P2*, and *P5*) displayed closer *Ho* and *He* values, suggesting the existence of subpopulation structures. The inbreeding coefficient (*F*) further illustrates this complexity, with positive and negative values across different populations and loci. The mean values of *F* over the five loci in each population range from −0.046 (HL_ZHFL) to 0.111 (HL_RRRS). These findings underscore the complex genetic structures within *F. taiwana* populations, characterized by varying levels of heterozygosity and inbreeding coefficients.

### 3.2. Regional Genetic Differentiation in F. taiwana across Taiwan’s Central Mountain Range by the Analysis of Molecular Variance

To assess the extent of genetic differentiation among *F. taiwana* populations across the central mountain ranges, we employed the Analysis of Molecular Variance (AMOVA) method [25]. We separated the eleven populations into three regions (eastern, midwestern, southwestern) for AMOVA analysis (Table 1). AMOVA results reveal the distribution of genetic variation at different hierarchical levels. Intriguingly, the majority of genetic variability (92%) is found within individuals, highlighting significant intra-individual genetic diversity. In contrast, variation among regions (eastern, midwestern, southwestern) contributes 5% to the total genetic variance, indicating a moderate, yet significant, geographic impact on genetic differentiation. Meanwhile, variation among populations within these regions is relatively minor (1%), pointing to extensive gene flow, or inherent genetic similarities, among these populations. The genetic variation between individuals within populations is about 2%, suggesting limited genetic differentiation at this level. The fixation indices, including differentiation among regions (*F_RT_* = 0.055, *p* = 0.001), among populations within regions (*F_SR_* = 0.01, *p* = 0.001), overall population differentiation (*F_ST_* = 0.064, *p* = 0.001), inbreeding within populations (*F_IS_* = 0.021, *p* = 0.08), and total genetic differentiation (*F_IT_* = 0.084, *p* = 0.001), collectively mirror these patterns of genetic variation. When the eleven *F. taiwana* populations were grouped into two regions, eastern and western, for AMOVA analysis, the variation among these regions accounted for 6%. The differentiation among regions (*F_RT_*) became 0.064 (*p* = 0.001), and the overall population differentiation (*F_ST_*) reached 0.075 (*p* = 0.001). The moderate *F_ST_* value indicates a noticeable level of genetic differentiation among populations, while the non-significant *F_IS_* value suggests that inbreeding is not a major concern within populations.

Further investigation into genetic differentiation and gene flow among *F. taiwana* populations was carried out by calculating pairwise *F_ST_* values (Table 4) and the number of migrants (*Nm*) (Table 5). The observed higher *F_ST_* values and lower *Nm* in Table 5 suggest pronounced genetic differentiation between populations from the western (TC_DKMZ, CH_HRHT, PT_MJ, and KH_BGZY) and the eastern (HL_BAXC, HL_DKXL, HL_GHJA, YL_NFSA, HL_RRRS, HL_TCGF, and HL_ZHFL) regions. This pattern indicates potential barriers to gene flow. A notable example is the *F_ST_* of 0.11 between TC_DKMZ (western) and HL_ZHFL (eastern), which signifies significant genetic differentiation. Conversely, genetic differentiation between the midwestern (TC_DKMZ and CH_HRHT) and the southwestern (PT_MJ, and KH_BGZY) regions is less pronounced, exemplified by an *F_ST_* of 0.026 between CH_HRHT (midwestern) and PT_MJ (southwestern). Within regions, the lower *F_ST_* values and higher *Nm*, as depicted in Table 5, indicate reduced genetic differentiation and increased gene flow among populations, likely due to their closer geographical proximity and similar environmental conditions.

These patterns of genetic differentiation and gene flow are further substantiated by the results of the hierarchical clustered heatmap and Principal Coordinate Analysis (PCoA), based on the *F_ST_* values. The heatmap (Figure 2A) distinctly reveals intra-regional genetic homogeneity within *F. taiwana* populations. In contrast, the PCoA (Figure 2B) demonstrates that the western populations form a distinct cluster, separate from the eastern populations, highlighting the clear genetic divergence between these two regions.

### 3.3. Bayesian Clustering Analysis of F. taiwana Unveiling Two Distinct Genetic Clusters across Taiwan’s Central Mountain Range

While AMOVA provided initial insights into the distribution of genetic variation across *F. taiwana* populations, we furthered our analysis using STRUCTURE software, a model-based Bayesian clustering analysis [20], to identify distinct genetic clusters within the eleven populations. The results revealed that the most likely number of genetic clusters (K) within the eleven *F. taiwana* populations was determined to be two. This was ascertained through the evaluation of the rate of change in the log probability of data between successive K values, which showed a clear peak at K = 2 (Figure 3A,B). In the STRUCTURE results, Cluster 1 predominantly comprised individuals from the western populations, while Cluster 2 mainly included individuals from the eastern populations (Figure 3C). This indicates that *F. taiwana* populations inhabiting the western region of Taiwan have a distinct genetic makeup compared to those in the eastern region. The admixture analysis revealed that most individuals from a specific population belonged predominantly to one cluster, reflecting a strong regional genetic structure, but some exhibited mixed ancestry. Notably, certain individuals from the KH_BGZY population in the western region displayed significant characteristics of Cluster 2 (Figure 3C), hinting at some level of gene flow between populations situated on opposite sides of the mountain ranges. This is likely attributable to historical migration or recent human-mediated movement. The *F_ST_* ranged from 0.12 to 0.29, confirming moderate genetic differentiation between the identified clusters. These STRUCTURE findings corroborate the AMOVA results, highlighting significant genetic variation among populations located on either side of the central mountain ranges. This genetic division is probably indicative of the ecological barrier imposed by the mountain ranges.

## 4. Discussion

Taiwan’s mountain ranges, comprising 268 peaks exceeding 3000 m in elevation, bisect the island into two distinct halves along a north–south axis. This geographic division creates a stark contrast between the rugged, steep terrain of eastern Taiwan and the gentler, rolling landscapes with expansive plains in the west. Such topographical diversity fosters varied ecosystems and communities across the eastern and western regions of the island. Previous studies have indicated genetic divergence in species inhabiting low-elevation areas on the eastern and western flanks of Taiwan [26,27]. In this study, we used microsatellite markers to genotype and provide the genetic evidence to demonstrate that the population structure of *F. taiwana* in western Taiwan is distinctly different from its counterpart in the eastern region.

In population genetics, the commonly utilized DNA markers can be developed from mitochondrial DNA (mtDNA) or nuclear genome. While mtDNA markers are relatively easy to develop, they come with several limitations: (1) only the maternal lineage can be traced, (2) mtDNA may be subject to selection pressures, (3) individuals may have multiple mtDNA haplotypes, and (4) mtDNA sequences can integrate into the nuclear genome, producing nonfunctional nuclear copies with sequence variations [28,29]. These factors can complicate the analysis and interpretation in mtDNA-based studies. In contrast, nuclear markers, such as microsatellite DNA, offer advantages like co-dominance, high polymorphism, Mendelian inheritance, and high variability in non-coding regions. Traditionally, developing microsatellite markers has been labor-intensive and expensive [30]. However, in this study, we developed microsatellite primers de novo and circumvented the traditional, laborious methods of microsatellite identification by employing PacBio long-read sequences. This approach, leveraging third-generation sequencing technology, makes microsatellite marker development more feasible and accessible.

The evaluation of observed heterozygosity (*Ho*), expected heterozygosity (*He*), and inbreeding coefficient (*F*) across eleven *F. taiwana* populations revealed a diverse genetic landscape. Some populations demonstrate high *Ho* and *He*, indicating substantial genetic diversity and effective random mating. However, discrepancies between *Ho* and *He* in other populations hint at possible inbreeding or population substructuring. We observed mixed *F* values within certain populations, with higher *F* values at specific loci, such as HL_RRRS (*F* = 0.386 for *P3* and *F* = 0.327 for *P4*), YL_NFSA (*F* = 0.228 for *P3*), HL_BAXC (*F* = 0.247 for *P3*), HL_TCGF (*F* = 0.237 for *P3*), and KH_BGZY (*F* = 0.286 for *P4*). Notably, this pattern was not uniform across all loci within the same populations (Table 3). This variability could be attributed to various factors, including subpopulation structures. Pesticide use, often employed to control *F. taiwana* populations, might induce subpopulation structures as genetic variants conferring resistance could become dominant. This shift could alter allele frequencies, potentially increasing *F* values. However, confirming this hypothesis requires further targeted research, taking into account both ecological and genetic factors. Despite observing mixed *F* values in some populations, Analysis of Molecular Variance (AMOVA) results suggest that inbreeding is not a predominant issue within these populations.

Analysis of pairwise *F_ST_* from AMOVA, along with the hierarchical clustered heatmap and Principal Coordinate Analysis (PCoA), revealed that *F. taiwana* populations in western Taiwan form a distinct cluster, distinctly separated from those in the eastern region. The *F_RT_* value (*F_RT_* = 0.064, *p* = 0.001) suggests moderate genetic differentiation between the eastern and western regions. This observation is further corroborated by model-based clustering analysis using STRUCTURE software, reinforcing the notion of clear genetic divergence between these two regions. Notably, some individuals from the KH_BGZY population in the western region exhibit genetic characteristics typical of eastern populations. KH_BGZY, being the Taiwan Native Botanical Garden, might have inadvertently caused the human-mediated dispersion of *F. taiwana*. This could have happened if larvae, which reside in the soil, were unintentionally transported during the transplantation of plants from the eastern region to the botanical garden. Such an event could introduce genetic material from the eastern population into the western population.

The central mountain ranges in Taiwan acts as a significant topographic barrier, potentially restricting the intermixing of populations on either side, and thus facilitating early stages of genetic differentiation despite ongoing gene flow. Given *F. taiwana*’s average flight height of 2 m and limited flying ability [8,10], active flight over the central mountain ranges is almost impossible. Furthermore, *F. taiwana*’s habitat is predominantly below 500 m in elevation, indicating difficulty in colonizing higher elevations. These factors suggest that the topographic barrier created by the central mountain ranges significantly restricts gene flow, as evidenced in this study. While passive dispersal mediated by human activities, such as transportation via vehicles, could contribute to *F. taiwana* dispersion, the marked genetic differentiation between western and eastern populations implies that this human-mediated movement, though present, may have limited effect. Additionally, Taiwan’s distinct monsoon winds, including the southwest monsoon in summer and the northeast monsoon in winter (Figure 1), could theoretically facilitate passive dispersion. However, considering their directional patterns, their contribution to gene flow between the eastern and western regions appears limited. Thus, we conclude that the gene flow of *F. taiwana* between these regions is primarily influenced by the topographic constraints imposed by Taiwan’s mountain ranges.

## Figures and Tables

**Figure 1 insects-15-00023-f001:**
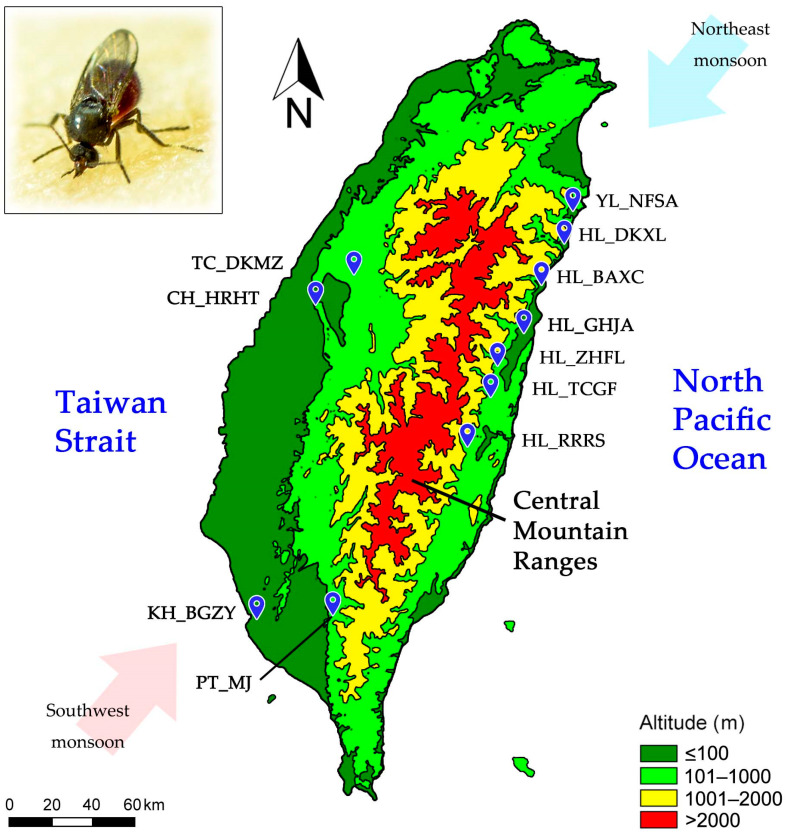
Sampling Locations of *F. taiwana* in Taiwan. The map illustrates the altitude of the landscape, with variations indicated by the color code. The wind directions of the southwest monsoon in summer (pink arrow) and the northeast monsoon in winter (blue arrow) are shown. The scale bar represents 60 km. Inset: A female *F. taiwana* biting human skin.

**Figure 2 insects-15-00023-f002:**
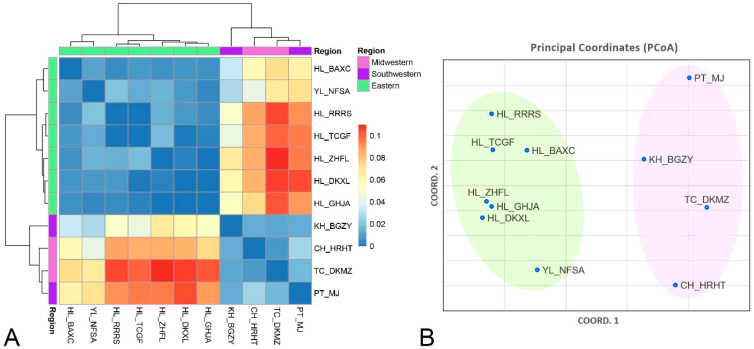
Genetic Differentiation of *F. taiwana* Populations between the Western and the Eastern Regions. (**A**) A hierarchical clustered heatmap, constructed based on *F_ST_* values, reveals a pronounced inter-regional genetic differentiation and a notable intra-regional genetic homogeneity among these populations. The heatmap includes a color code for geographical regions on the right, along with a color bar representing the range of *F_ST_* values. (**B**) The Principal Coordinate Analysis (PCoA), utilizing these *F_ST_* values, demonstrates that the western populations (TC_DKMZ, CH_HRHT, PT_MJ, and KH_BGZY) form a distinct cluster, separate from the eastern populations (HL_BAXC, HL_DKXL, HL_GHJA, YL_NFSA, HL_RRRS, HL_TCGF, HL_ZHFL). The x-axis (COORD.1) accounts for 80% of the observed variation, while the y-axis (COORD.2) accounts for 8.84%.

**Figure 3 insects-15-00023-f003:**
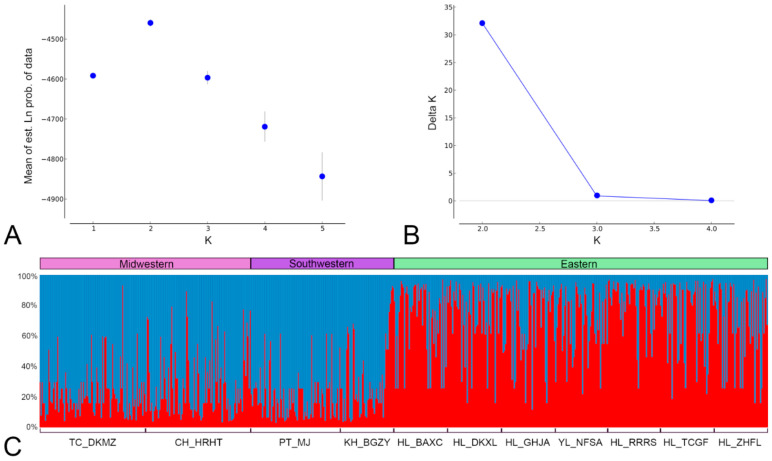
Bayesian Clustering Analysis of *F. taiwana* Using STRUCTURE Reveals Two Distinct Genetic Clusters. The optimal number of genetic clusters was determined using the ΔK method, as implemented on the Structure Harvester website. (**A**) Displays the mean estimated log probability of the data for different assumed numbers of genetic clusters (K), ranging from 1 to 5. The likelihood of the data peaks at K = 2, indicating this as the optimal number of genetic clusters. (**B**) Shows the Delta K values, calculated based on the rate of change in log probability between successive K values, with a pronounced peak at K = 2, further supporting the existence of two distinct genetic clusters. (**C**) Illustrates the percentage stacked bar chart of aligned clustering results for K = 2, generated using Clumpp software. Each individual is represented by a vertical bar partitioned into colored segments indicating their estimated membership in the two clusters. Cluster 1 mainly includes individuals from western populations (TC_DKMZ, CH_HRHT, PT_MJ, and KH_BGZY), while Cluster 2 consists primarily of individuals from eastern populations (HL_BAXC, HL_DKXL, HL_GHJA, YL_NFSA, HL_RRRS, HL_TCGF, and HL_ZHFL). Populations are delineated along the x-axis, with regional labels provided above the chart.

**Table 1 insects-15-00023-t001:** Geographic Locations and Sample Sizes of *F. taiwana* Specimens.

Geographic Location in Taiwan (Code)	Geographic Region	Latitude	Longitude	Altitude (Meters)	Sample Size
Nan’ao Farm, Su’ao Township, Yilan County (YL_NFSA)	Eastern	24.453	121.790	18	47
Taiwan Cement DAKA Park, Xiulin Township, Hualien County (HL_DKXL)	Eastern	24.300	121.751	13	48
Bo’ai Road, Xincheng Township, Hualien County (HL_BAXC)	Eastern	24.126	121.651	18	47
Guanghua 10th Street, Ji’an Township, Hualien County (HL_GHJA)	Eastern	23.932	121.558	54	48
Zhonghua Road, Fenglin Town, Hualien County (HL_ZHFL)	Eastern	23.747	121.451	100	48
Tangchang Street, Guangfu Township, Hualien County (HL_TCGF)	Eastern	23.658	121.420	121	48
Ruisui Ranch, Ruisui Township, Hualien County (HL_RRRS)	Eastern	23.479	121.345	154	47
Dakeng Scenic Area, Minzheng Li, Taichung City (TC_DKMZ)	Midwestern	24.172	120.752	205	95
Hushan Rock, Huatan Township, Changhua County (CH_HRHT)	Midwestern	24.043	120.559	56	94
Majia Township, Pingtung Country (PT_MJ)	Southwestern	22.703	120.648	228	81
Native Botanical Garden, Zuoying District, Kaohsiung City (KH_BGZY)	Southwestern	22.680	120.301	9	48

**Table 2 insects-15-00023-t002:** Characteristics of the Five Microsatellite Loci of *F. taiwana* Used in This Study.

Locus	Size Range (bps)	Motif Type ^1^	Primer Sequences ^2^ and the Concentration of the Primers Used in the PCR Reactions	Number of Alleles	Accession Number
*P1*	138~183	(TAA)_7_	F: TCAGGCTCGTACGGTTTACT (200 nM)	7	MW921469
			R: GGAAGTCAAAATAGGAGTTTCTCAA (200 nM)		
*P2*	213~239	(ATT)_8_	F: AAATCAATAAATATTACGAGTTTTCCA (100 nM)	8	MW921470
			R: TTTTTATCGTTCAAAATCCTTCTG (200 nM)		
*P3*	256~264	(TGA)_6_	F: GGGGCCTCCTCATAAGTACC (100 nM)	5	MW921471
			R: TGCATTTCTTGCACTCCATT (200 nM)		
*P4*	348~359	(CAA)_6_ + (CCA)_6_	F: CCCACGCACTAACGAGAGTT (150 nM)	5	MW921472
			R: AGCGGGAGTTCCATTTCTCT (200 nM)		
*P5*	453~465	(TTC)_7_	F: TCCTCACCCCAAAAAGTGAC (100 nM)	6	MW921473
			R: AGGGAGCGACTTCAAATCAA (200 nM)		

^1^ The number of repeats for microsatellite markers in the PCR-amplified sequences deposited in GenBank is indicated; ^2^ All of the forward (F) primers have an *M13* primer sequence (CACGACGTTGTAAAACGAC) extension at their 5′ ends.

**Table 3 insects-15-00023-t003:** Genetic Structure of *F. taiwana* Across Eleven Populations and Five Microsatellite Loci.

Population	Locus	N	*Na*	*Ne*	*Ho*	*He*	*uHe*	*F*
TC_DKMZ	*P1*	95	4.000	2.306	0.589	0.566	0.569	−0.041
	*P2*	95	4.000	1.137	0.126	0.120	0.121	−0.048
	*P3*	95	2.000	1.134	0.126	0.118	0.119	−0.067
	*P4*	95	4.000	1.367	0.263	0.268	0.270	0.019
	*P5*	95	4.000	1.149	0.116	0.130	0.130	0.107
CH_HRHT	*P1*	94	4.000	2.223	0.596	0.550	0.553	−0.083
	*P2*	94	5.000	1.242	0.160	0.195	0.196	0.181
	*P3*	94	5.000	1.229	0.181	0.186	0.187	0.028
	*P4*	94	4.000	1.570	0.383	0.363	0.365	−0.055
	*P5*	94	4.000	1.366	0.245	0.268	0.269	0.087
PT_MJ	*P1*	80	3.000	1.666	0.363	0.400	0.402	0.094
	*P2*	81	4.000	1.190	0.173	0.160	0.161	−0.082
	*P3*	81	2.000	1.261	0.210	0.207	0.208	−0.014
	*P4*	81	3.000	1.412	0.272	0.292	0.293	0.069
	*P5*	81	3.000	1.191	0.173	0.160	0.161	−0.077
KH_BGZY	*P1*	48	3.000	2.251	0.500	0.556	0.562	0.100
	*P2*	48	5.000	1.549	0.333	0.355	0.358	0.060
	*P3*	48	3.000	1.317	0.271	0.241	0.243	−0.125
	*P4*	48	3.000	1.539	0.250	0.350	0.354	0.286
	*P5*	48	4.000	1.186	0.167	0.157	0.159	−0.062
HL_BAXC	*P1*	47	5.000	1.958	0.511	0.489	0.495	−0.043
	*P2*	47	7.000	2.345	0.532	0.574	0.580	0.073
	*P3*	47	2.000	1.394	0.213	0.282	0.286	0.247
	*P4*	47	5.000	1.246	0.170	0.198	0.200	0.139
	*P5*	47	2.000	1.394	0.255	0.282	0.286	0.096
HL_DKXL	*P1*	48	6.000	2.313	0.667	0.568	0.574	−0.174
	*P2*	48	7.000	2.956	0.688	0.662	0.669	−0.039
	*P3*	48	2.000	1.110	0.104	0.099	0.100	−0.055
	*P4*	48	4.000	1.135	0.104	0.119	0.120	0.126
	*P5*	48	3.000	1.567	0.333	0.362	0.366	0.079
HL_GHJA	*P1*	48	5.000	1.945	0.500	0.486	0.491	−0.029
	*P2*	48	7.000	2.855	0.646	0.650	0.657	0.006
	*P3*	48	2.000	1.087	0.083	0.080	0.081	−0.043
	*P4*	48	3.000	1.290	0.208	0.225	0.227	0.072
	*P5*	48	3.000	1.640	0.333	0.390	0.395	0.146
YL_NFSA	*P1*	47	5.000	2.392	0.574	0.582	0.588	0.013
	*P2*	47	6.000	2.295	0.596	0.564	0.570	−0.056
	*P3*	47	2.000	1.160	0.106	0.138	0.139	0.228
	*P4*	47	4.000	1.508	0.298	0.337	0.341	0.116
	*P5*	47	3.000	1.655	0.447	0.396	0.400	−0.129
HL_RRRS	*P1*	47	5.000	2.179	0.638	0.541	0.547	−0.180
	*P2*	46	6.000	2.635	0.587	0.621	0.627	0.054
	*P3*	47	2.000	1.209	0.106	0.173	0.175	0.386
	*P4*	47	4.000	1.462	0.213	0.316	0.319	0.327
	*P5*	47	2.000	1.367	0.277	0.268	0.271	−0.031
HL_TCGF	*P1*	48	5.000	2.383	0.583	0.580	0.586	−0.005
	*P2*	48	7.000	3.130	0.708	0.681	0.688	−0.041
	*P3*	48	3.000	1.158	0.104	0.137	0.138	0.237
	*P4*	48	4.000	1.383	0.271	0.277	0.280	0.023
	*P5*	48	2.000	1.358	0.271	0.264	0.266	−0.027
HL_ZHFL	*P1*	48	6.000	1.801	0.458	0.445	0.450	−0.030
	*P2*	48	6.000	2.122	0.625	0.529	0.534	−0.182
	*P3*	48	2.000	1.180	0.167	0.153	0.154	−0.091
	*P4*	48	3.000	1.135	0.125	0.119	0.120	−0.051
	*P5*	48	2.000	1.679	0.354	0.404	0.409	0.124

N: Number of successfully genotyped samples; *Na*: Number of different alleles; *Ne*: Number of effective alleles; *Ho*: Observed heterozygosity; *He*: Expected heterozygosity; *uHe*: Unbiased heterozygosity: (2N/(2N − 1)) × *He*; *F*: Inbreeding coefficient: (*He* − *Ho*)/*He*.

**Table 4 insects-15-00023-t004:** Pairwise *F_ST_* Values and Statistical Significance for *F. taiwana* Populations. *F_ST_* values are presented below the diagonal, while *p*-values, derived from 999 permutations, are shown above the diagonal.

	TC_DKMZ	CH_HRHT	PT_MJ	KH_BGZY	HL_BAXC	HL_DKXL	HL_GHJA	YL_NFSA	HL_RRRS	HL_TCGF	HL_ZHFL
**TC_DKMZ**	0.000	0.014	0.004	0.020	0.001	0.001	0.001	0.001	0.001	0.001	0.001
**CH_HRHT**	0.009	0.000	0.001	0.014	0.001	0.001	0.001	0.001	0.001	0.001	0.001
**PT_MJ**	0.017	0.026	0.000	0.014	0.001	0.001	0.001	0.001	0.001	0.001	0.001
**KH_BGZY**	0.012	0.014	0.014	0.000	0.001	0.001	0.001	0.001	0.001	0.001	0.001
**HL_BAXC**	0.073	0.057	0.062	0.034	0.000	0.065	0.090	0.028	0.105	0.068	0.109
**HL_DKXL**	0.106	0.083	0.103	0.058	0.008	0.000	0.437	0.062	0.053	0.295	0.196
**HL_GHJA**	0.101	0.074	0.089	0.053	0.007	0.000	0.000	0.141	0.103	0.179	0.396
**YL_NFSA**	0.068	0.043	0.072	0.028	0.011	0.010	0.005	0.000	0.003	0.011	0.021
**HL_RRRS**	0.105	0.086	0.091	0.052	0.006	0.009	0.006	0.020	0.000	0.413	0.018
**HL_TCGF**	0.100	0.084	0.095	0.044	0.009	0.002	0.004	0.014	0.000	0.000	0.010
**HL_ZHFL**	0.110	0.084	0.095	0.066	0.007	0.004	0.000	0.016	0.015	0.018	0.000

**Table 5 insects-15-00023-t005:** Pairwise Comparison of the Number of migrants (*Nm*) Among *F. taiwana* Populations. *Nm*, calculated using the formula *Nm* = ((1/*F_ST_*) − 1)/4, is shown below the diagonal. n.a. (not applicable) indicates cases where *F_ST_* = 0.

	TC_DKMZ	CH_HRHT	PT_MJ	KH_BGZY	HL_BAXC	HL_DKXL	HL_GHJA	YL_NFSA	HL_RRRS	HL_TCGF	HL_ZHFL
**TC_DKMZ**	0.000										
**CH_HRHT**	27.571	0.000									
**PT_MJ**	14.584	9.387	0.000								
**KH_BGZY**	19.947	18.185	17.746	0.000							
**HL_BAXC**	3.186	4.099	3.780	7.172	0.000						
**HL_DKXL**	2.108	2.762	2.166	4.039	29.808	0.000					
**HL_GHJA**	2.218	3.124	2.565	4.459	36.629	n.a.	0.000				
**YL_NFSA**	3.444	5.517	3.246	8.617	21.595	25.792	48.282	0.000			
**HL_RRRS**	2.129	2.642	2.502	4.574	38.291	27.387	38.790	12.212	0.000		
**HL_TCGF**	2.254	2.711	2.383	5.370	28.100	119.469	61.551	18.122	n.a.	0.000	
**HL_ZHFL**	2.027	2.717	2.384	3.520	37.650	63.288	n.a.	15.118	16.698	13.920	0.000

## Data Availability

The data presented in this study are available on request from the corresponding author.

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
