# Peer review of "Genetic Differentiation of the Bloodsucking Midge Forcipomyia taiwana (Diptera: Ceratopogonidae): Implication of the Geographic Isolation by the Central Mountain Ranges in Taiwan"

_insects, 2024, doi:10.3390/insects15010023_

Round 1

Reviewer 1 Report

Comments and Suggestions for Authors

Very well written manuscript. Just a few minor editorial suggestions provided in the attached file. In my opinion, the authors clearly demonstrate that geneflow between populations of Forcipmyia taiwana is affected by the central mountain ranges in Taiwan. My biggest point of criticism on the manuscript is the sampling regime for the different populations of F. taiwana. I am concerned that where the populations were sampled might artificially lead to some of the genetic clusters observed. For example, if more populations were sampled in southeastern and southern Taiwan (operating on the assumption that F. taiwana occurs around the periphery of Taiwan), would those populations cluster closer to the other eastern populations (HL populations) or closer to southwestern populations (KH and PT). However, this sampling concern does not detract from the main conclusion of the manuscript, which is to show that the central mountain ranges affects gene flow in this species. 

Reviewer 2 Report

Comments and Suggestions for Authors

This is a straightforward, simple study of the population genetic structure within a pest midge from Taiwan. Several analyses are performed on exclusively microsatellite data and the authors conclude that the central mountain ranges in Taiwan are causing moderate isolation between eastern and western populations on the island (e.g. FST = 0.075), a not entirely surprising result. The authors appear to focus on the pest status of this insect in order to make it seem more interesting.

Generally, this is well-written and reasonably conducted. However, the interpretation of the STRUCTURE analyses needs to be qualified more and there is one questionable statement that is prominent in the justification/utility of this study (see below).

“These findings provide crucial insights for understanding the species' molecular ecology and can inform targeted pest management strategies.”

It is not clear at all how the knowledge gained in this study would inform pest management.  This is stated twice in the manuscript and never explained. If this statement is to be made as a justification/deliverable for this study (it’s the final sentence in the abstract) then it must be expanded upon in the introduction and/or discussion sections. There is already considerable discussion of the pest nature of the insect, so it could go into that section. If there is not really a defensible justification for the above statement, it should be taken out, as it’s hand-waving.

The STRUCTURE analysis does not show a particularly clear break between eastern and western populations. At least not as clear as the authors seem to indicate. The western (southwestern) population KH_BGZY has several individuals that exhibit eastern population assignment. This should be discussed.

Minor error:

Figure 1 shows a topographical map of the island and the legend in the lower right corner says “Altitude (km)”. It should say “m”, not “km”.
